# TokenLearner: Adaptive Space-Time Tokenization for Videos

**Michael S. Ryoo**[1,2]**, AJ Piergiovanni**[1]**, Anurag Arnab**[1]**, Mostafa Dehghani**[1]**, Anelia Angelova**[1]
[1]Google Research
[2]Stony Brook University
{mryoo,ajpiergi,aarnab,dehghani,anelia}@google.com

## Abstract

In this paper, we introduce a novel visual representation learning which relies on a handful of adaptively learned tokens, and which is applicable to both image and video understanding tasks. Instead of relying on hand-designed splitting strategies to obtain visual tokens and processing a large number of densely sampled patches for attention, our approach learns to mine important tokens in visual data. This results in efficiently and effectively finding a few important visual tokens and enables modeling of pairwise attention between such tokens, over a longer temporal horizon for videos, or the spatial content in image frames. Our experiments demonstrate strong performance on several challenging benchmarks for video recognition tasks. Importantly, due to our tokens being adaptive, we accomplish competitive results at significantly reduced computational cost. We establish new state-of-the-arts on multiple video datasets, including Kinetics-400, Kinetics-600, Charades, and AViD.

The code will be available at: `https://github.com/google-research/scenic/tree/main/scenic/projects/token_learner`

## 1 Introduction

Videos provide an abundance of visual information. Video understanding particularly requires employing effective spatial-temporal processing of frames to capture long-range interactions [5, 37, 21, 17, 24, 12, 34, 20, 25, 1]. An important aspect of this understanding is how to quickly learn which parts of the input video stream are important, both spatially and temporally, and to focus computational resources on them. But what basic processing mechanism are able to do so successfully?

Recent advancements in Transformers demonstrate improved accuracy on vision classification tasks. For example, departing from standard convolutional approaches, the Vision Transformer (ViT) [9] treats the image as a sequence of patches, utilizing the Transformer architecture [39] similar to text understanding. Standard approaches for video recognition take videos as stacked images (i.e., a space-time volume) and tend to extend 2D neural architectures to 3D (e.g., 3D-ResNets [17, 5, 38, 11]). Motivated by ViT, recent approaches [2, 3] also extend Transformers for videos by creating 3D 'tubelet' video tokens with regular 3D-grids, which often result in computationally heavy models. There are often too many tokens to process, especially for longer videos.

The main question addressed in this work is how to adaptively learn the representation from visual inputs to most effectively capture the spatial information for image frames and spatio-temporal interactions for videos. Here are our main ideas:

The first key observation is we are able to learn to represent visual data by learning to 'tokenize' the representations. This is in contrast to previous approaches which used densely sampled tokens e.g., 16x16 or 32x32 over a series of attention layers [9, 3].

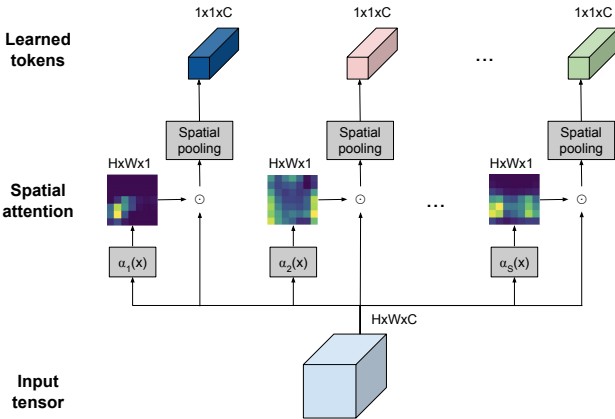

Figure 1: Visual illustration of the TokenLearner module, applied to a single image frame. TokenLearner learns to spatially attend over a subset of tensor pixels (i.e., from intermediate spatial representations), and generates a set of token vectors adaptive to the input.

Specifically, we can learn to compute important regions in the input image/video, making the tokens adapt to the input data. We compute multiple spatial weight maps per frame with a spatial attention mechanism, and use it for the tokenization. The goal of these maps is to learn which areas are of importance. Here, each spatial weight map is multiplied with the input to form a 'token', to be processed by the subsequent learning modules.

Furthermore, we find that very few tokens may be sufficient for a visual understanding task. More specifically, we show that one can significantly reduce the computational budget of video Transformers, by utilizing 8-16 tokens as an intermediate frame representation (instead of keeping 200∼500). Our TokenLearner is able to reduce the number of total FLOPS by half, while maintaining or even increasing the classification accuracy.

The approach is simple, efficient, and, as shown by the results, outperforms methods including both convolutional methods and previous space-time Transformer ones from prior art. In video understanding tasks, we establish new state-of-the-art numbers on Kinetics-400, Kinetics-600, Charades, and AViD datasets by outperforming prior models.

## 2   TokenLearner Modules for Adaptive Tokenization

In visual Transformer architectures such as ViT [9], an input image is first tokenized by splitting it into small (e.g., 16x16) spatial patches, which are used as input to the model. Similarly, in recent video Transformer architectures, such as ViViT [2] and TimeSformer [3], the video is tokenized by cutting the video into 2d spatial or 3d spatio-temporal cubes on a regular grid.

Instead of processing fixed, tokenized inputs, our attention module learns the tokens that are to be used for the recognition task. We gain several important properties by doing so: (1) We enable the adaptive tokenization so that the tokens can be dynamically selected conditioned on the input. (2) This also effectively reduces the total number of tokens for the transformer, which is particularly beneficial considering that there are many tokens in videos (e.g., $14 \times 14 \times 64$) and the computation is quadratic to the number of tokens. (3) Finally, we provide an ability for each subsequent layer to learn to rely on different space-time tokenizations, potentially allowing different layers to capture different aspects of the video. These dynamically and adaptively generated tokens can be used in standard transformer architectures such as ViT for images and ViViT for videos.

### 2.1   TokenLearner

Let $X$ be an input tensor with a space-time shape: $X \in \mathbb{R}^{T \times H \times W \times C}$ where $H \times W$ corresponds to the spatial dimension of the input, $T$ is the temporal dimension (i.e., number of frames), and $C$ is the number of channels. Let $X_t$ be a temporal slice of it, corresponding to the frame $t$: $X_t \in \mathbb{R}^{H \times W \times C}$.

In the case of an image input, $T = 1$ and $X = X_t$. Note that $X$ could also be an intermediate representation within a network, and $X_t$ will be its slice in such case.

For every time frame $t$, we learn to generate a series of $S$ tokens, $Z_t = [z_i]_{i=1}^S$, from the input frame $X_t$. Specifically, we formulate a tokenizer function, $z_i = A_i(X_t)$, which maps the input frame $X_t$ to a token vector $z_i$: $\mathbb{R}^{H \times W \times C} \mapsto \mathbb{R}^C$. The idea is to learn our tokenizer function $A_i$ to adaptively select an informative combination of pixels (or spatial locations) in $X_t$, and we have $S$ number of such functions. This way, our tokens will not be fixed splits of the input tensor, but a set of adaptively changing spatial selections. Different tokens will be mined per frame, allowing us to model their space-time relations/interactions in case of videos. We also set $S$ to be smaller than $H \times W$ (e.g., $S = 8$ and $H \times W = 14 \times 14$), enabling the model to significantly reduce the computations needed for the layers following this module.

Here, our tokenizer $z_i = A_i(X_t)$ is implemented with a spatial attention mechanism: i.e., the model learns to compute a weight map (of size $H \times W$) conditioned on the input $X_t$, and is multiplied with $X_t$ itself. More specifically, let $\alpha_i(X_t)$ be a function generating the spatial $H \times W \times 1$ weight map. Each token $z_i$ is generated by

$$z_i = A_i(X_t) = \rho(X_t \odot A_{iw}) = \rho(X_t \odot \gamma(\alpha_i(X_t))),  \tag{1}$$

where $\odot$ is the Hadamard product (i.e., element-wise multiplication) and $A_{iw} \in \mathbb{R}^{H \times W \times C}$ is an intermediate weight tensor computed with the function $\alpha_i(X_t)$ and the broadcasting function $\gamma(\cdot)$. Finally, spatial global average pooling $\rho(\cdot)$ is applied on top of them to reduce the dimensionality to $\mathbb{R}^C$. The resulting tokens are gathered to form the output tensor: $Z_t = [z_i]_{i=1}^S \in \mathbb{R}^{S \times C}$.

The overall process has a form of an element-wise spatial self-attention. In our version, $\{\alpha_i(\cdot)\}_{i=1}^S$ are implemented together as a single or a series of convolutional layers (with the channel size $S$) followed by a sigmoid function, although this could be extended with other implementations. In case of an image, $Z = Z_t$. In the case of a video, the tokens $Z_t$ from all the frames are collected to form the final output token tensor $Z \in \mathbb{R}^{ST \times C}$.

We specifically name our token learning module as "TokenLeaner". Figure 1 visually summarizes the TokenLearner module.

**Compute reduction in Transformers:**  The learned tokens (i.e., the outputs of the TokenLearner $Z$) are provided to the subsequent layers for the visual representation learning, such as multi-head self-attention (MHSA) used in Vision Transformer and ViViT. With the TokenLearner, these subsequent layers only need to process a small number of tokens (e.g., 8 instead of 1024 per frame) and this significantly reduces the computations, as they are quadratic to the number of tokens. Figure 4 (a) shows a basic architecture inserting the TokenLearner module within ViViT. It could be added at any location within the network, and the relative compute of the Transformer layers after the TokenLearner become almost negligible due to the huge difference in the number of tokens.

## 2.2 TokenFuser

After the TokenLearner generates tokens and its subsequent Transformer layer (e.g., MHSA) processes them, the "TokenFuser" could be used to further (1) fuse information across the tokens and (2) remap the representation back to its original spatial resolution. This enables the model to capture spatial (or spatio-temporal) 'patterns' formulated by the tokens, and recover the original input tensor shape when necessary.

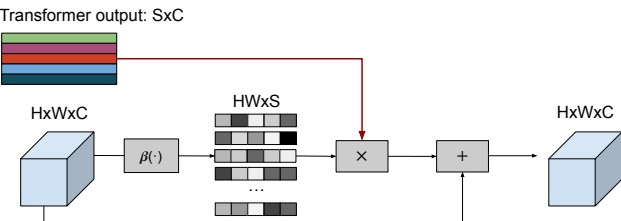

Figure 2: Visual illustration of the TokenFuser module, applied to each image frame individually.

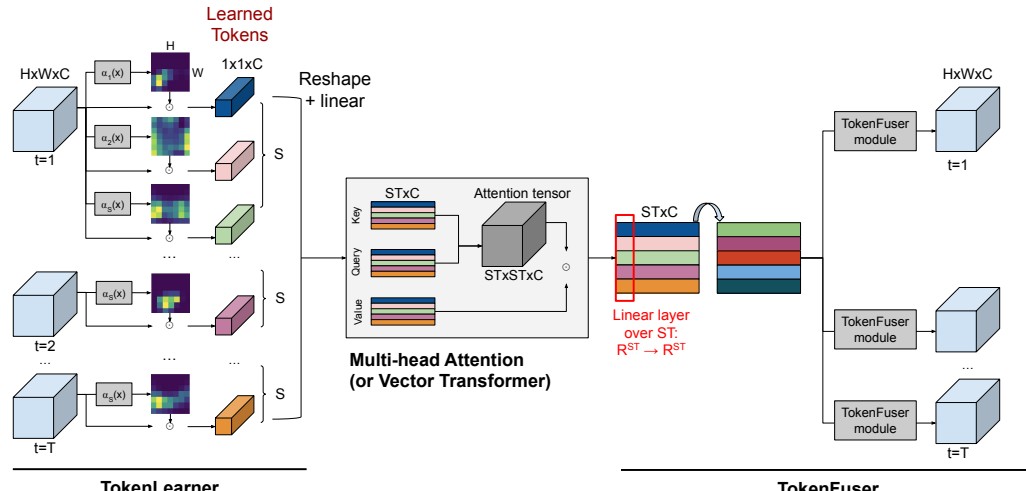

Figure 3: TokenLearner, Transformer, and TokenFuser combined for video representation learning. TokenLearner first learns to generate a set of token vectors, Transformer (e.g., MHSA) models their space-time relations, and TokenFuser combines them. $S$ is the number of tokens we learn per frame, and $T$ is the number of frames. Note that this combination can serve as a 'module' itself, and one may stack such module multiple times within the network. TokenFuser could be dropped.

First, given the token tensor $Y \in \mathbb{R}^{ST \times C}$ from a Transformer layer, we apply a linear layer (i.e., a fully connected MLP layer) over the tokens, not channels. That is, we learn a linear function of $\mathbb{R}^{ST} \mapsto \mathbb{R}^{ST}$ where $S$ is the number of our tokens mined per frame and $T$ is temporal size of the input tensor, and apply it to every channel independently. That is, we update $Y = (Y^T M)^T$ where $M$ is a learnable weight matrix with size $ST \times ST$. The result of such operation maintains the tensor size of $ST \times C$. We believe this also has a connection to the observations from the concurrent work, MLPMixer [36], that token-wise linear layers are beneficial.

Next, the TokenFuser processes each temporal slice $Y_t \in \mathbb{R}^{S \times C}$ individually, and remaps the token tensor of size $S \times C$ back to $H \times W \times C$, by learning to combine the tokens for each spatial location in $H \times W$ differently.

$$X_t^{j+1} = B(Y_t, X_t^j) = B_w Y_t + X_t^j = \beta_i(X_t^j)Y_t + X_t^j \tag{2}$$

where $X_t^j$ is the residual input to the previous TokenLearner module, $Y_t$ is the processed tokens in the TokenFuser module, and $X_t^{j+1}$ is the output. $B_w \in \mathbb{R}^{HW \times S}$ is an intermediate weight tensor computed with the function $\beta_i(X_t)$. The function $\beta_i(X_t)$ is implemented with a simple linear layer followed by a sigmoid function.

Figure 2 illustrates the overall process of the TokenFuser (the token-wise linear layer is omitted).

## 2.3 Video architecture overview

Here, we provide an overview of video representation architecture with TokenLearner. The TokenLearner and TokenFuser modules introduced in Section 2 are directly applicable for video representation learning. TokenLearner generates multiple $Z_t$ for frames in videos and they are stacked to form $Z$. Once $Z$ is generated, any standard Transformer layers could be used to parse them jointly.

Figure 3 provides an overview of the combined architecture for video representation, which is to be repeated over multiple layers. TokenLearner first extracts $S$ number of tokens per frame, resulting in a total of $ST$ tokens where $T$ is the number of frames. Once TokenLearner generates these adaptively learned tokens, they are provided to the subsequent Transformer layer to capture the global space-time patterns. Finally (and optionally depending on the architecture), TokenFuser applies a linear layer over the token axis and remaps the tensor shape back, as discussed in Subsection 2.2. Following Eq. 2, TokenFuser is applied for per-frame representation $Y_t$. This results in a lightweight approach, which brings forth an efficient video representation by capturing long-range visual patterns.

# 3 Experiments: TokenLearner with Video Vision Transformer

## 3.1 Network architecture implementation

In this experiment, we use the Video Vision Transformer (ViViT) architecture [2], following its detailed settings and implementation [7]. ViViT is a direct extension of ViT [9] for videos, which uses spatio-temporal tubelets from videos as its tokens. The size of the space-time tubelets are typically 16x16x2, which are given to the Transformer layers.

We use ViViT-L/16 as our backbone, while also applying the TokenLearner to backbones with more initial tokens such as L/14 and L/10. ViViT-L models have 24 transformer layers. Following the setting of [2], we used the input resolution of 224x224, extracting tubelets, and attaching positional encodings.

Figure 4 (a) and (b) show two different architectures incorporating TokenLearner. (a) is formed by inserting TokenLearner in the middle of the network such as after the 12th layer among 24, while (b) uses both TokenLearner and TokenFuser. In particular, our model (b) is formed by replacing conventional Transformer layers with a series of TokenLearner-Transformer-TokenFuser. Similar to (a), such replacement is done only for the layers after a certain point. For instance, we keep twelve of the standard Transformer MHSA layers in the beginning, and replaces the remaining twelve layers with our TokenLearner-Transformer-TokenFuser modules repeated twelve times. We also modified L/14 and L/10 models to have more transformer layers (e.g., 35 instead of 24). Note that the computation increase caused by the transformer layers added after TokenLearner module is relatively very small, as the number of tokens are few: 8 or 16 per frame.

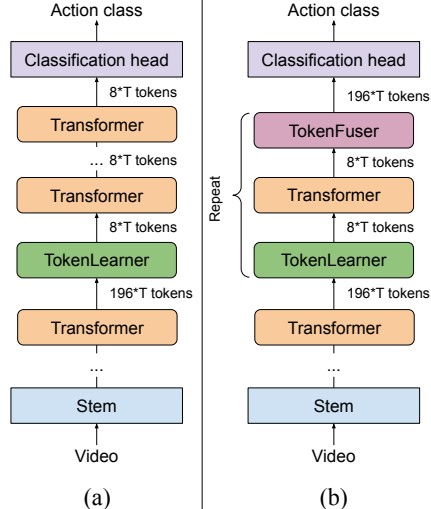

Figure 4: Our models following the ViViT architecture. (a) with TokenLearner and (b) with both TokenLearner and TokenFuser.

We tried various number of tokens including $S = 8, 16, 32$, and use $S = 8$ and 16 as our default settings. That is, the TokenLearner is learning to abstract an image frame into 8 (or 16) tokens. The spatial attention function $(\alpha)$ in TokenLearner is implemented with four 3x3 conv. layers (with gelu in between), whose channel size is identical to the number of tokens (e.g., $S = 8$).

## 3.2 Datasets and training

We use the Kinetics datasets, which are video classification datasets with relatively short video clips (∼10 seconds). We train and evaluate on both Kinetics-400 and Kinetics-600 datasets, which have about 240k and 390k training samples. We follow the standard settings used in previous papers and report accuracy on the validation set [5, 12].

Following ViViT [2], we first pretrain models on JFT [35] to obtain initial weights. The weights of the initial convolutional layers to handle image patches (e.g., 16x16) are processed to handle 16x16x2 video patches by following ViViT's 3D initialization strategy, and the weights of the Transformer and the TokenLearner layers are directly inherited.

## 3.3 Results

We evaluate various versions of the ViT-L models incorporating the TokenLearner module. As mentioned above, all of the models are pre-trained on JFT and finetuned on Kinetics. We use the standard L/16 models + TokenLearner, as well as L/14 and L/10. L/14 and L/10 use 11 additional layers compared to the standard ViT L/16, but as also described in the above subsections, the computation increase caused by them are minimal due to the number of tokens being much smaller, 8 or 16 per frame, in the added layers. We report both their classification accuracies and FLOPS.

Table 1 compares the accuracies of the base ViViT models against our ViViT + TokenLearner models on Kinetics-400. These models are directly comparable as they follow the exact same setting and

Table 1: Comparison of ViViT models with and without TokenLearner on Kinetics-400. GLOPS are per view. The difference in the number of parameters between the TokenLearner models comes from the different number of layers used after the TokenLearner module.

| Method | Top-1 accuracy | Top-5 accuracy | # params. | GFLOPS |
|---|---|---|---|---|
| ViViT-L/16 [2] | 82.8 | 95.5 | 308M | 1446 |
| ViViT-L/16 320 [2] | 83.5 | 95.5 | 308M | 3992 |
| ViViT-H/14 [2] | 84.8 | 95.8 | 654M | 3981 |
| ViViT-L/16 (our run) | 83.4 | 95.6 | 308M | 1446 |
| TokenLearner 16at12 + L/16 | 83.5 | 95.6 | 308M | 766 |
| TokenLearner 8at18 + L/16 | 84.5 | 96.1 | 383M | 1105 |
| TokenLearner 16at18+ L/14 | 84.7 | 96.1 | 447M | 1621 |
| TokenLearner 16at18+ L/10 | 85.4 | 96.3 | 450M | 4076 |

Table 2: ViViT + TokenLearner on Kinetics-400, compared to the state-of-the-art models. Different approaches rely on different pre-training datasets, such as ImageNet-21K (for TimeSformer and Swin) and JFT (for ViViT and TokenLearner). The multiplication in GFLOPS correponds to the number of views used for the inference, such as 4x3 = 12.

| Method | Top-1 accuracy | Total GFLOPS |
|---|---|---|
| R(2+1)D [38] | 73.9 | $304 \times 115$ |
| SlowFast 16x8, R101+NL [12] | 79.8 | $234 \times 30$ |
| TimeSformer-L [3] | 80.7 | $2380 \times 3$ |
| ViViT-L/16 [2] | 82.8 | $1446 \times 12$ |
| ViViT-H/14 [2] | 84.8 | $3981 \times 12$ |
| Swin-L [23] | 83.1 | $604 \times 12$ |
| Swin-L (384) [23] | 84.6 | $2107 \times 12$ |
| Swin-L (384) [23] | 84.9 | $2107 \times 50$ |
| TokenLearner 16at12 (L/16) | 82.1 | $766 \times 6$ |
| TokenLearner 8at18 (L/16) | 83.2 | $1105 \times 6$ |
| TokenLearner 16at12 (L/16) | 83.5 | $766 \times 12$ |
| TokenLearner 8at18 (L/16) | 84.5 | $1105 \times 12$ |
| TokenLearner 16at18 (L/14) | 84.7 | $1621 \times 12$ |
| TokenLearner 16at18 (L/10) | **85.4** | $4076 \times 12$ |

the pre-train dataset. "TokenLearner 16at12" means that we have the TokenLearner layer learning 16 tokens, after the 12th Transformer layer. We are able to observe that the use of TokenLearner enables better classification while also reducing the compute. In particular, inserting TokenLearner in the middle of the network (at 12) achieves better accuracy than the base mode, while cutting the computation by (almost) half. In addition, having the TokenLearner at the later layer (at 18) achieves even superior accuracy while still performing faster, thanks to its adaptiveness.

Table 2 compares the TokenLearner accuracy against the state-of-the-arts models. Note that these approaches follow slightly different settings and pretrain datasets (e.g., the use of ImageNet-21K instead of JFT like ours). We believe the accuracy of 85.4 is the highest that has been reported so far, and we believe it is meaningful. Table 3 compares the results on Kinetics-600. Similar to our results on Kinetics-400, we are able to observe that our proposed approach extends the state-of-the-arts while also being computationally efficient.

## 4 Experiments: TokenLearner with Bottleneck Transformer

### 4.1 Network architecture implementation

In this experiment, we follow the Bottleneck Transformer [33] network style, while taking advantage of X3D [11] as the backbone. This is motivated by the successful usage of X3D on Charades.

Table 3: ViViT + TokenLearner on Kinetics-600. The multiplication in GFLOPS correponds to the number of views used for the inference, such as 4x3 = 12.

| Method | Top-1 | Total GFLOPS |
|---|---|---|
| SlowFast 16x8, R101+NL [12] | 81.8 | $234 \times 30$ |
| X3D-XL [11] | 81.9 | $48 \times 30$ |
| TimeSformer-HR [3] | 82.4 | $1703 \times 3$ |
| ViViT-L/16 [2] | 84.3 | $1446 \times 12$ |
| ViViT-H/14 [2] | 85.8 | $3981 \times 12$ |
| Swin-B [23] | 84.0 | $282 \times 12$ |
| Swin-L (384) [23] | 85.9 | $2107 \times 12$ |
| Swin-L (384) [23] | 86.1 | $2107 \times 50$ |
| TokenLearner 16at12 (L/16) | 84.4 | $766 \times 12$ |
| TokenLearner 8at18 (L/16) | 86.0 | $1105 \times 12$ |
| TokenLearner 16at18 (L/10) | 86.1 | $4076 \times 12$ |
| TokenLearner 16at18 w. Fuser (L/10) | **86.3** | $4100 \times 12$ |

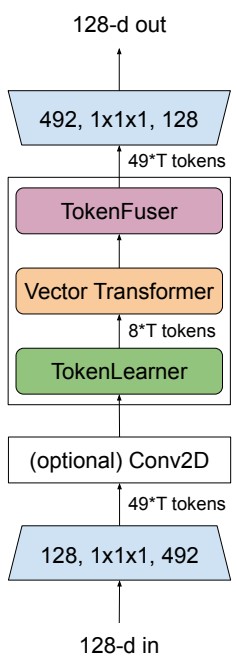

Figure 5: Our network module following the bottleneck transformer, with X(2+1)D backbone. It is an inverted bottleneck.

Specifically, we modified X3D to be more computationally efficient by (1) replacing its 3D XYT convolutional layers with a pair of 2D conv. layer and 1D conv. layer, and (2) removing Squeeze-and-Excitation layers [18] and swish activations. Our backbone could be viewed as X(2+1)D. We use the channel sizes and the number of layers identical to X3D-M, which is an efficient model.

Based on such X(2+1)D architecture, and following the Bottleneck Transformer concept, we replace the space-time convolution layers in the last block with our transformers. Figure 5 illustrates the residual module architecture, which is repeated multiple times in the block. TokenLearner, Transformer, TokenFuser are applied in a sequence, with an optional 2D $3 \times 3$ convolution layer before them. The spatial attention function (i.e., $\alpha(\cdot)$) in TokenLearner is implemented with a single conv2d layer.

Here, we used a Vector Transformer instead of MHSA as our Transformer layer, which could be also viewed as the MHSA with the number of heads being identical to the number of channels. We provide more details in Appendix.

We use $224 \times 224 \times 64$ videos for training and $256 \times 256 \times 64$ videos for testing. After the 3rd residual block, the input tensor has the shape of $8 \times 8 \times 64$, and this becomes the input to the TokenLearner. For an efficient implementation the intermediate channel size of TokenLearner was set identical to the output channel size, $d = 432$. Notice that 64 frames were used to best capture longer-term temporal information. $S = 8$ number of tokens were used.

### 4.1.1 Datasets

**Charades dataset**: The Charades dataset [31] is a dataset collected by assigning activity tasks which people in various environments are acting out, by performing a sequence of actions which involve interaction with objects. For example, sitting on the couch and reading a book, closing the book, standing up and speaking on the phone. It comprises 8000 training and 1686 validation videos with an average duration of 30 seconds. It has 157 activity classes. This dataset is very challenging as it is a multi-class, multi-label video dataset, that is, more than one activity can occur at the same time, and it includes fine grained motions or interactions with small objects in real-world environments. We follow the standard evaluation protocols, reporting the mean Average Precision (mAP) % (v1 classification setting of the dataset). We used the frame rate of 6 fps and 12 fps to obtain the training/testing videos. The dataset has a Non-Commercial Use license.

Table 4: Performance on the Charades multi-label classification task. 12 fps setting. Performance is measured using the Mean Average Precision (mAP) since more than one ground truth action is possible. Methods with RGB and optical flow input modalities are listed.

| Method | Input | Pre-train | mAP |
|---|---|---|---|
| I3D [5] | RGB | Kinetics | 32.9 |
| I3D from [40] | RGB | Kinetics | 35.5 |
| I3D + Non-local [40] | RGB | Kinetics | 37.5 |
| EvaNet [26] | RGB | Kinetics | 38.1 |
| STRG [41] | RGB | Kinetics | 39.7 |
| LFB-101 [43] | RGB | Kinetics | 42.5 |
| SGFB-101 [19] | RGB | Kinetics | 44.3 |
| SlowFast-101 [12] | RGB+RGB | Kinetics | 45.2 |
| AssembleNet-50 [30] | RGB+Flow | None | 47.0 |
| Multiscale ViT [10] | RGB | Kinetics | 47.7 |
| AssembleNet-101 [30] | RGB+Flow | Kinetics | 58.6 |
| AssembleNet++ [29] (w/o object) | RGB+Flow | None | 55.0 |
| MoViNets [22] | RGB | None | 63.2 |
| Backbone (X(2+1)D-M) | RGB | None | 62.7 |
| Ours | RGB | None | **66.3** |

Table 5: Performance on the Anonymized Videos from Diverse countries (AViD) dataset. Performance in terms of mean accuracy is shown in % averaged over 887 classes. Previous approaches results are reported from [27], all based on training from scratch with RGB-only inputs.

| Method | Accuracy | total GFLOPS |
|---|---|---|
| I3D [5] | 46.5 | $108 \times$ N/A |
| (2+1)D ResNet-50 | 46.7 | $152 \times 115$ |
| 3D ResNet-50 | 47.9 | N/A |
| SlowFast-50 8x8 [12] | 50.2 | $65.7 \times 30$ |
| SlowFast-101 16x4 [12] | 50.8 | $213 \times 30$ |
| Backbone (X(2+1)D-M) | 48.6 | $532 \times 1$ |
| X(2+1)D-M w/ disjoint space+time Transformer (like [3]) | 50.6 | $493 \times 1$ |
| Ours | **53.8** | $487 \times 1$ |

**AViD dataset**: The Anonymized Videos from Diverse countries (AViD) dataset [27] is a unique dataset which is representative of the world's population video content generation. It is collected from videos uploaded from multiple countries across six continents and demonstrates higher diversity compared to other video datasets such as Kinetics in its concepts, actions and visual representations. For example a 'greeting' in certain countries involves a handshake, in some a kiss, but in others a slight bow. The dataset is explicitly designed to contain less bias, encourage diversity, while respecting privacy and licenses. The AViD dataset contains 887 classes and 450k videos (410k training 40k testing) and is of comparable size to Kinetics-400 and Kinetics-600 datasets with 400 and 600 classes respectively, also containing variable duration videos $3 - 15s$. We report classification accuracy over the 887 classes. All the videos in this dataset have the Creative Commons License.

## 4.2 Results

**Charades dataset results:** In Table 4 we compare the proposed TokenLearner to the state-of-the-art methods. Our approach outperforms these, including several recent works. The mAP of 66.3% on Charades classification establishes the new state-of-the-art.

**AViD results:** Table 5 shows the results on the AViD dataset. As seen, our approach outperforms prior work on this challenging dataset too. We also compared ours to the reimplementation of TimeSformer module [3] applied to the same backbone as ours. This uses disjoint spatial and temporal transformer modules, which was also tested in [2]. We are able to observe that we establish the new state-of-the-arts on this dataset, while also being more computationally efficient.

Table 6: Comparison between TokenLearner and the joint space-time transformer modules similar to [2], applied to our backbone. They use the X(2+1)D backbone, tested on Charades with the 6 fps setting, Charades 12 fps setting, and AViD dataset. GFLOPs and # params are of each module (with 64 frame inputs), not the entire network.

| Module | Char-6fps | Char-12fps | AViD | GFLOPs | # params |
|---|---|---|---|---|---|
| Joint space-time MHSA | 57.9 | 64.0 | 53.3 | 22.0 | 0.30M |
| Conv2D + Joint space-time MHSA | 58.6 | 62.5 | 52.5 | 35.8 | 1.98M |
| Ours (TokenLearner) | 58.8 | 63.4 | **53.8** | 3.4 | 0.81M |
| Ours (Conv2D + TokenLearner) | **59.6** | **66.3** | 53.7 | 17.2 | 2.49M |

### 4.3 Ablations

**Comparison against different tokenizations:** Here, we compare the model with TokenLearner against space-time transformer modules with the standard tokenization. More specifically, we compare the use of TokenLearner + Vector Transformer + TokenFuser against the full joint space-time transformer module (advocated in [2] and also mentioned in [3]), without token learning. The full joint space-time transformer module is a transformer layer on space-time tokens similar to ours, but it relies only on the hand-designed tokenization. Compared to TokenLearner which generates $S \times T$ number of tokens, the full joint space-time transformer uses $H \times W \times T$ number of tokens. In our bottleneck implementation, it uses $\sim 8$ times more tokens (i.e., 8*64 vs. 8*8*64). For the joint space-time transformer modules, the standard multi-head self-attention (MHSA) with 8 heads is used.

Table 6 shows the results. Interestingly, despite the heavier computation of the full joint space-time transformer, it performed slightly worse to the TokenLearner modules. We believe this shows the advantage of the 'adaptiveness' of the tokens in the TokenLearner and shows that the standard transformers might be suffering from the tokens irrelevant to the actions serving as noise or distractors.

We also report the amount of computation and the number of parameters of each module in these models. This depends on the input size and the hyper parameter setting, and our measurement is based on the input size (i.e., $T \times H \times W \times C$) of $8 \times 8 \times 64 \times 492$. Note that this is the measurement of modules, not the entire network.

**Comparison between multiple space-time layer combinations.** As also suggested in previous literature, it is a common strategy for video representations to pair a layer focusing on spatial information with a layer focusing on temporal information (e.g., R(2+1)D [38] and TimeSformer [3]). Table 7 shows the results of this ablation. For spatial and temporal transformer implementations, the standard multi-head self-attention was used, as was done in [2, 3]. The result shows that the proposed TokenLearner is more accurate than other popular combinations. The modules based on TokenLearner also effectively only uses a fraction of the Tokens per frame (i.e., 8) as opposed to other methods which use $16 \times 16$ or $32 \times 32$ tokens.

One of the main benefits of the TokenLearner (in addition to the adaptive tokenization of the input and that we explicitly fuse the tokens to capture their spatio-temporal patterns) is that, unlike the disjoint space/time transformers used in this ablation study, it is a joint space-time transformer. Simultaneously, it still manages its computation to be much more tractable (as shown in Tables 6 and 7): A naive full version of the space-time transformer would require consideration of $8 \times 8 \times 64 = 4096$ tokens in our case, building and multiply the attention tensor of size $4096 \times 4096$. On the other hand, the TokenLearner learns to consider $8 \times 64 = 512$ tokens jointly.

**More TokenLearner alternatives.** We also compared our spatial attention-based token learning with alternative approaches: (1) using a fixed grid to split each frame into the same number of tokens (i.e., 8 tokens), (2) the approach of directly generating tokens using a fully connected layer, and (3) the approach of spatially average pooling the entire frame pixels and using fully connected layers to generate multiple tokens per frame. In the second approach, we directly model $z_i = A_i(x)$ as a dense layer, producing $T \times S \times C$ tensor based on the $T \times H \times W \times C$ input. The third approach is similar, except that we apply spatial global average pooling per frame and then use MLP to generate tokens.

Table 7: Comparison between different space-time transformer modules. They were all applied to the same backbone architecture (i.e., the Bottleneck Transformer-style with X(2+1)D). The Charades-6fps is used in this experiment. FLOPS are estimated with 64-frame settings, per module.

| Module | Charades-6fps (%) | GFLOPs | # params |
|---|---|---|---|
| Conv2D + Conv1D | 56.6 | 18.3 | 2.24M |
| Conv2D + MLPMixer [36] | 57.0 | 13.8 | 2.06M |
| Conv2D + Temporal transformer | 58.4 | 16.5 | 1.98M |
| Spatial + Temporal transformer | 58.8 | 5.5 | 0.59M |
| Conv2D + Spatial + Temporal transformer | 58.0 | 19.2 | 2.27M |
| Ours (TokenLearner) | 58.8 | 3.4 | 0.81M |
| Ours (SpatialT + TokenLearner) | 58.9 | 6.2 | 1.11M |
| Ours (Conv2D + TokenLearner) | 59.6 | 17.2 | 2.49M |

The fixed split tokenization method (1) provided us the accuracy of 58.8 on Charades, as opposed to 59.6 of ours. The direct token generation method (2) provided the accuracy of 56.6 on Charades, failing to obtain better tokens. Pooling and generation method (3) gave us the accuracy of 58.6. These results suggest the importance of spatial attention for the token learning, our TokenLearner. The same vector transformer and TokenFuser (from Section 2) were used for this ablation.

## 5 Related work

Video understanding relies on both the spatial and the temporal information in the video. In order to adequately capture both motion and appearance information in videos, full 3D space-time convolutional layers as well as (2+1)D convolutional layers have been used [37, 5, 38, 44]. More advanced network designs have also been extremely popular in video CNNs particularly two-stream ones [32, 13, 14, 15, 8, 12] and, recently, architecture searched ones [11, 30, 26].

Attention-based architectures, e.g., the Transformer [39] have shown remarkable success in both Natural Language processing (NLP) and computer vision. Most adaptations of the Transformer architectures to computer vision, have been slow, although some optimizations, have been successful e.g., for image classification, [4, 45, 6, 28] and for video generation [42].

Applying attention-based architectures to video presents a definite challenge as the model needs to learn dependencies across both the spatial and temporal domains. The Vision Transformer [9] demonstrated how the NLP-specific Transformer architecture can elegantly work for images, by subdividing the input image into non-overlapping patches on a regular grid and feeding them as token embeddings to the Trasnformer, where $O(N^2)$ tokens are used or order of 256 or 1024. [16] relied on the region proposal network to use the detected human and object candidates as tokens, showing that it could be combined with CNNs.

A couple of recent work [2, 3], in the spirit of the Vision Transformer, subdivided the video into token in a 3D grid to capture the video input. This leads to $O(N^3)$ increase in the number of tokens required for learning (typically $\sim$ 25k tokens for 96-frame model). Our work, in contrast, learns the tokens from data which results in a significantly fewer tokens, and more efficient approach. We see that even 8x times fewer tokens (e.g., 512 vs 4096), when learned, are able to capture successfully the information needed for video representation learning.

## 6 Conclusions

We have presented TokenLearner, a novel approach for visual representation learning, which adaptively tokenizes the representations. The goal is to learn to extract important tokens in image frames and videos for the recognition tasks at hand. Our approach is more efficient, than contemporary work, by finding few important space-time tokens which can model visual representations of images and videos. We observe improved accuracies across challenging video understanding tasks, and outperformed prior approaches in many datasets. One of the remaining challenges is in learning full spatio-temporal tokens. The current TokenLearner focuses on finding spatial tokens over a sequence of frames, and it could be extended to directly mine tokens over space-time volumes.

## Acknowledgement

We thank Dmitry Kalashnikov, Andy Zeng, and Robotics at Google NYC team members for valuable discussions on attention mechanisms.

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
