# TokenLearner: Adaptive Space-Time Tokenization for Videos
# - Supplementary Materials -

**Michael S. Ryoo**[1,2], **AJ Piergiovanni**[1], **Anurag Arnab**[1], **Mostafa Dehghani**[1], **Anelia Angelova**[1]
[1]Google Research
[2]Stony Brook University
{mryoo,ajpiergi,aarnab,dehghani,anelia}@google.com

## A  Appendix

### A.1  Vector Transformer: Pairwise vector attention

Here, we summarize the details of the Vector Transformer used in the Bottleneck Transformer experiments.

Once TokenLearner generates adaptively learned tokens, a vector attention between key-query pairs could be computed. This can be thought as a version of multi-head self-attention in which the number of heads is the same as channels, allowing us to learn a different attention matrix for each channel. It captures in an efficient way pairwise space-time relations per channel, particularly benefiting tokens with rich channel information.

Given $Z$, a set of tokens reflecting different space-time aspects of a video, the Transformer models space-time interactions between them. In particular, we follow the formulation of [5], which enables a vector-version of the Transformer, although it is also possible to incorporate other attention mechanisms.

For every token $z_i$, the output of the Transformer $y_i$ is computed by considering all possible $z_j$ as:

$$y_i = \sum_{z_j \in Z} \gamma(f_q(z_i) \odot f_k(z_j)) \odot f_v(z_j) \tag{1}$$

where $i$ and $j$ are the indexes of the tokens in $Z$ whose size is $|Z| = ST$. $f_q$, $f_k$, and $f_v$ are the linear layers projecting the tokens. $\gamma$ is an extra projection layer to match the channel dimensionality followed by a softmax function over $j$. When the channel sizes of the projections are identical, $\gamma$ is simplified as a single softmax layer identical to the standard transformer.

In the original transformer notation, the query matrix $Q$ corresponds to our $\{f_q(z_i)\}_i$, and the key matrix $K$ corresponds to our $\{f_k(z_j)\}_j$. Instead of computing the dot product between $Q$ and $K$ as $QK^T$ to generate the attention 'matrix', this vector formulation computes an attention 'tensor' $\{\gamma(f_q(z_i) \odot f_k(z_j))\}_{(i,j)}$ preserving the channel information. It has shape $ST \times ST \times d$ where $d$ is the intermediate channel size. The computed attention tensor is multiplied with the value matrix $\{f_v(z_j)\}_j$ to get the final transformer outputs.

Notice that this vector transformer is a global representation, and the temporal range of the information it is able to capture entirely depends on what tokens we provide to it. With our learnable adaptive tokens, we have the capability to cover a larger number of frames and focus on the temporal structure.

### A.2  Video classification training details

**ViViT**   We follow the exact training protocols and the hyper parameters of [1]. We use the same code (the Scenic library [2]) and the hardware for the training as well as for the evaluation.

35th Conference on Neural Information Processing Systems (NeurIPS 2021).

Table 1: TokenLearner compared against pooling-based token reduction.

| Details | ImageNet | GFLOPS |
|---|---|---|
| Base ViT L/16 | 87.35 | 363.1 |
| 2x2 pool at 9 and 18 | 85.63 | 144.3 |
| 2x2 pool at 12 and 18 | 86.41 | 187.2 |
| 4x4 pool at 12 | 83.93 | 184.4 |
| TokenLearner 16at12 | 87.68 | 184.6 |

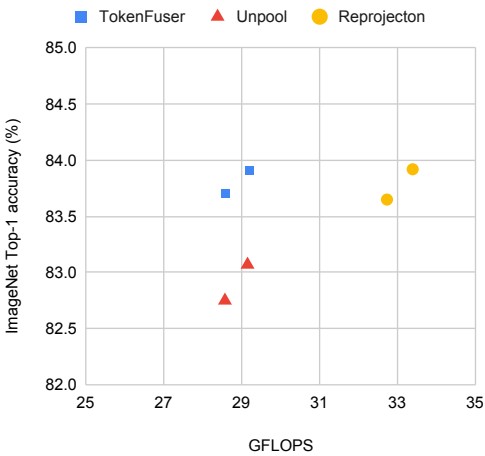

Figure 1: Ablations with TokenFuser alternatives.

We train the Kinetics model for 30 epochs with the base learning rate of 0.05 with the Momentum optimizer. Basically, all the settings in our Kinetics experiments follow the setting of ViViT.

**Bottleneck Transformer**   We provide the training details as below. For the training/testing splits of the datasets, we followed their standard settings.

We use the cosine-decay learning rate which was popularly used in many video CNN model trainings. The base learning rate of 0.8 per TPU core (which is equivalent to a single GPU) is used for the Charades dataset (multi-label action classification) and the base rate of 0.025 per TPU was used for the AViD dataset (video classification). The training was done for 100k iterations with the batch size of 4 per TPU core (i.e., 4*64=256 was our batch size) in the Charades experiments. The batch size of 8 per TPU core was used for AViD. 100k iterations correspond to roughly 125 epoches in AViD. Label smoothing of 0.2 was used for the AViD training. No label smoothing was used for the Charades. In Charades, the training was done by temporally cropping a long Charades videos (e.g., 30 seconds) into 64 frame segments. The evaluation was done similarly with 64 frame segments by merging their output responses.

The training time of a single model was around ∼16 hours with 32 TPU v3. This was bottlenecked by the data pipeline, and the actual computation is less.

## A.3   Additional ablations

**TokenLearner vs. pooling.**   A straightforward alternative to the TokenLearner module is the use of spatial pooling to reduce the number of tokens. It can be done by spatially rearranging the tokens to have the height and width, and then applying conventional spatial pooling. This is similar to the pooling-based MHSA module used in [3].

Table 1 compares the TokenLearner against the spatial pooling, when we add TokenLearner to ViT for image classification on ImageNet (pretrained on JFT). The architecture described in Figure 4 (a) of the main paper was adopted. In all these experiments, ViT L/16 model was used. We are able to observe that there is a benefit in token 'learning'. The pooling-based token reduction does have computation similar to the TokenLearner, but it loses its accuracy compared to the base model. On the other hand, TokenLearner performs a bit better than the base model despite the low computation.

**TokenFuser alternatives.**   Here, we experimentally compare the proposed TokenFuser module with its alternatives. The role of the TokenFuser is to mix the output tokens from the Transformer layer and map it back to the original shape before the token reduction.

The most straightforward alternative would be to (1) use the masks from the TokenLearner module to 'unpool' the output tokens. The idea is to multiply each output token with the corresponding spatial map computed during the previous TokenLearner module, and sum all of them to recover the original

Table 2: Comparing different components of our TokenLearner. On Charades dataset (6fps).

| Module | Accuracy (%) |
|---|---|
| Standard transformer (MHSA) | 58.4 |
| Vector transformer (VectT) | 58.1 |
| Prior-only-attention + broadcasting | 58.6 |
| Vector transformer (VectT) + broadcasting | 58.9 |
| Vector transformer (VectT) + TokenFuser | 59.0 |
| TokenLearner + MHSA + TokenFuser | 59.0 |
| TokenLearner + VectT + TokenFuser | 59.6 |

input tensor shape. Alternatively, (2) we can use one more transformer layer to increase the number of tokens back to the original number of tokens, similar to the 're-projection' used in [4].

Figure 1 shows the results with B/16, on ImageNet classification. The architecture described in Figure 4 (a) of the main paper was adopted. The unpooling strategy performes worse. The re-projection strategy performs comparably to the TokenFuser, but requires more FLOPS.

**Different components.** Using the setting of the Bottleneck Transformer experiments, we did an ablation to evaluate components of our approach and their combinations. We conducted ablations removing/adding different components of our model. In addition to Vector Transformer described in the above subsection, we also tried an ablation of replacing it with the multi-head self-attention. Table 2 shows the results, demonstrating the benefits each of the elements bring to the approach. For this experiment, we used the module composed of Conv2D + transformer (within the bottleneck), which we found to perform the best from the other ablations.

## A.4 Visualizations

Figure 2 shows visualizations of the tokens being learned with our approach. We show the spatial attention maps (i.e., $\alpha_i(x)$) from the first TokenLearner module, as the inputs to the higher-level TokenLearner becomes more mixed spatially and temporally. We are able to observe that they tend to focus more on human regions, and that they change over time responding to the changes in the visual input. Among the $S = 8$ tokens per frame we learn, we visualize 4 of them.

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

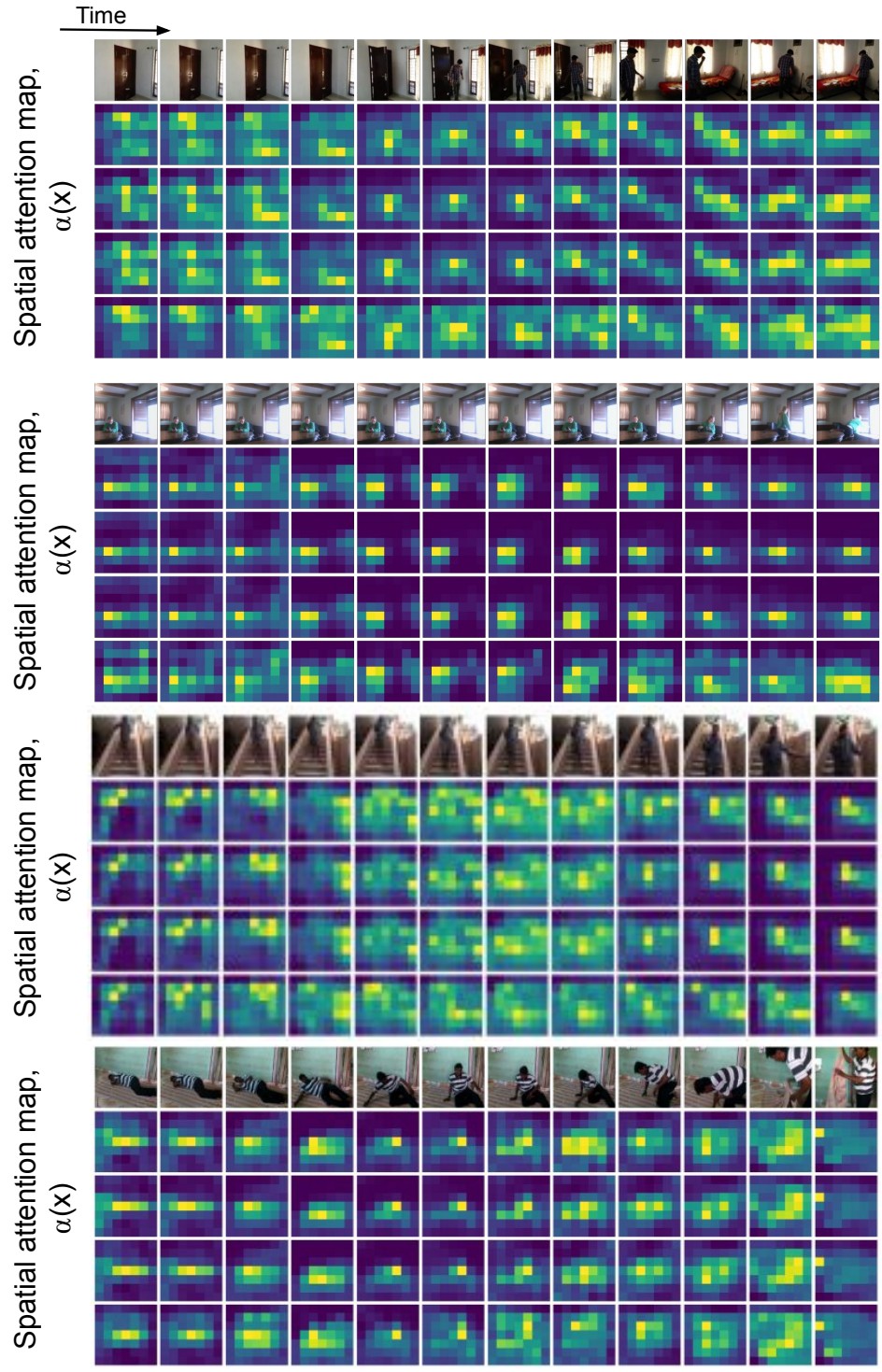

Figure 2: Visualization of the spatial attention maps for the tokenizations. Attention maps for four among a total of eight learned tokens are shown.