# OpenReview forum: "TokenLearner: Adaptive Space-Time Tokenization for Videos"
_NeurIPS.cc/2021/Conference — NeurIPS 2021 Poster_

### Official Review · Reviewer_h2mC · 2021-07-14

**Rating:** 6
**Confidence:** 4

**Summary:**

In this paper, the authors present an adaptive space-time tokenization for video representation learning. In the proposed method, each frame in a video is tokenized into a set of visual tokens through weighted summation over spatial locations. Then, vector transformer operates on a collection of tokens from entire video frames to temporally aggregate the information. The updated tokens are added back to the input tensor after linear pooling and broadcasting. The authors validated the proposed network module (TokenLearner) on two video action recognition benchmarks.

**Limitations And Societal Impact:**

I suggest the authors to discuss the limitations in the paper.


**Main Review:**

Strength:
- The idea of tokenizing frames instead of splitting into regular grids looks interesting and working well. The motivation is clear and the proposed approach is potentially beneficial to efficiency and performance of video recognition.
- The paper contains comprehensive ablation studies that compares the proposed approach with possible alternatives. Visualization of tokenization is also provided.
- In general, the paper is written clearly and easy to follow.

Weakness:
- I have a little concern about the technical novelty. Learning an adaptive tokenization of an image is not new. In [A], a very similar mechanism for tokenizing an image into a set of visual tokens are proposed. While this work is proposed for image representation learning, extending it to temporal domain is straightforward. The difference is, in the previous work, they project back the information in the tokens after applying transformer. Projecting the information into the right location looks better than the global update by broadcasting. The authors should discuss this paper.
- The experiments are weak. The authors use Charades and AViD dataset for experiments, which are not very popular in comparison with Kinetics and something-something. As a result, the proposed approach could not compared with recent transformer-based approaches. Comparing the proposed transformer-based method to only non-transformer-based methods is not convincing in an age when transformer-based methods are breaking new records every day. I strongly suggest the authors to include comparisons to recent transformer-based methods in more datasets.
- Impact of the number of tokens (S). S seems an important hyper parameters in the proposed method that makes trade-offs between speed and accuracy. It would be great if we can see its effects and the robustness on the choice of S.

Overall Evaluation:
- The paper proposed a tokenization approach for video representation learning. And, this is beneficial for video networks in terms of both speed and accuracy. But, I have some concerns on the technical novelty and the experiments, as discussed in Weakness. My current recommendation is on borderline slightly leaning toward accept. I would like to finalize my rating based on how the authors address my concerns during rebuttal.

**Time Spent Reviewing:**

3

---

> ### Author Response · Authors · 2021-08-10
> **Thank you for the comments. Please find our answers below.**
>
> ### Kinetics
>
> Thank you, following the suggestion from the reviewers, we evaluated TokenLearner on Kinetics400. This was done by inserting TokenLearner within ViViT, which is a full space-time Transformer architecture. Please refer to our response to Reviewer Lkvt: "Comparison against ViViT and the use of Kinetics" for the detailed results with tables.
>
> A couple of observations include:
> - The basic TokenLearner + ViViT L/16 performs better than ViViT L/16, while only spending ~1/2 FLOPS.
> - TokenLearner + L/10 obtains 85.4 classification accuracy, which to our knowledge is the best reported accuracy on this dataset.
>
> ### Comparison against [A]
>
> As the reviewer suggested, the main differences between [A] (which we believe is https://arxiv.org/pdf/2006.03677.pdf) and TokenLearner are :
>
> - TokenLearner enables video representation learning, and we demonstrate more broader usage of it within general transformer architectures such as ViViT and Bottleneck Transformer.
> - We have TokenFuser instead of the projection layer in [A]. The advantage of the TokenFuser is that it has a capability to learn patterns formed by tokens (i.e., spatio-temporal patterns across tokens) using a fully connected layer with a transposed axis. We found such global space-time pattern learning to be beneficial: we tried an approach similar to the projection layer that spatially remaps tokens to the input resolution, but this was not as beneficial as TokenFuser.
>
> Thank you - we will revise the paper to include this discussion.
>
> ### Impact of the number of tokens?
>
> Yes, thank you for the suggestion. Naturally more tokens have the ability to convey more information, whereas in the paper we specifically tried to keep the number of tokens fixed and small (8 and 16), in order to explore what is possible for the challenging task of video understanding with so few (adaptively learned) tokens.

---

### Official Review · Reviewer_Lkvt · 2021-07-14

**Rating:** 6
**Confidence:** 4

**Summary:**

The paper introduces a dynamic tokenizer that constructs the tokens using a set of convolutional layers, each selecting regions of interest. This is then combine with a space-time like attention mechanism. The idea is intersecting but important experiments are missing.

**Limitations And Societal Impact:**

The paper does a reasonable job to address this, perhaps a small comment: The conclusion lists "One of the remaining challenges [...]", what will be the others?

**Main Review:**

On novelty:
The idea of generating tokens dynamically makes sense. There is really no particular good reason why one should allocate one token per patch irrespective of its content. This is why I found it surprising that the paper doesn't actually test this in a transformer, as in replacing the tokenizer from ViViT (the 3D tokenizer) and TimeSformer/ViT (2d tokenizer) with this. As is, this is a variation of a spatio-temporal attention layer that is thrown in the network which makes the novelty more incremental, considering both views may solidify this work.

Results and ablation:
- To follow up on my previous point what is the performance of ViVit and Timesformer when their tokenizer is replaced with the proposed one? (Note, that the FLOPs should ideally be ketp alike to avoid bias)
- What is the accuracy of a network that exclusively uses this blocks ones (i.e. as in replacing the
- The authors mention multiple times ViViT and TimeSformer - yet there is no actual tete-a-tete comparison against them.
- Experiments on Kinetics are really a must if one want's to abe able to easily compare against the entirety of the state-of-the-art spectrum. Currently it's hard to properly understand how this approach positions itself in terms of accuracy in respect to the broader literature.
- What is the computational cost of the methods listed in Table 1&2?
-  L191 claims that this is a full space-time attention, motivating as such the reason why it outperforms disjoint space and time attention. Yet, in the supplementary material the claims are slightly different as the later actually outperforms the former. This is claimed to be due adaptivness. Given that a full space-time attention looks at all patches across space-time shouldn't this adapt too? Could the reason be due to overfitting of the space-time attention? This won't be too surprising as transformers already need much heavier augmentations to train well on images. As such, to what extend this results could be affected due to a) suboptiomal hyper-parameters and b)sub-optimal augmentations that may put to advantage one case over the other?


Others:
- The provided training details are insufficient to reproduce this work. What are the augmentations used for example?

- It was a bit unclear at first that the token learner is not intended to actually replace the tokenizer from transformers as portrayed early on in Section 2 when a parallel with ViViT and TimeSformer is made.

- The paper is only 8 pages long (out the  9 pages limit), as such I am not sure why there was a need to create an appendix instead of attaching it to the main paper as most ot the results will fit well in the main paper.

- L358 section number is missing

**Time Spent Reviewing:**

7

---

> ### Author Response · Authors · 2021-08-10
> **Thank you for the comments. Please find our answers below.**
>
> ### Comparison against ViViT and the use of Kinetics
>
> We thank the reviewer for these suggestions.
>
> Following them, we conducted additional experiments by inserting our TokenLearner directly into ViViT. We did this experiment on Kinetics400, allowing direct comparisons to previous work.
>
> We followed the same setting used in ViViT [2], using the same augmentation for the training and evaluating them with 4x3 views for the inference while using 224x224 input resolution and 32 frames. We kept the initial ViViT tokenization (i.e., spatio-temporal tubelets of size 16x16x2), and also experimented with smaller tubelets which would normally require too much memory to train, but are enabled by TokenLearner. We used ViViT-L which has a total of 24 transformer layers.
>
> The table below shows the results.
>
> \\begin{array} {l|c|c}
> \\text{Method} & \\text{Accuracy} & \\text{GFLOPS per view} \\\\
> \\hline
> \\text{ViViT-L/16 [2]} & 82.8 & 1446 \\\\
> \\text{ViViT-L/16 320 [2]} & 83.5 & 3992 \\\\
> \\text{ViViT-H/16 [2]} & 84.8 & 8316 \\\\
> \\hline
> \\text{ViViT-L/16 (our run)} & 83.4 & 1446 \\\\
> \\text{TL 16at12 + L/16} & 83.5 & 766 \\\\
> \\text{TL 8at18 + L/16} & 84.5 & 1105 \\\\
> \\text{TL 16at18+ L/14} & 84.7 & 1621 \\\\
> \\text{TL 16at18+ L/10} & 85.4 & 4076 \\\\
> \\end{array}
>
> We applied TokenLearner in the middle of ViViT where sufficient semantics could be captured in the TokenLearner module. "TL 16at12" means that we are inserting TokenLearner before the 12th transformer layer, with the number of tokens being 16. We kept the number of tokens (i.e., 8 or 16 per time step) identical in the layers after the TokenLearner insertion.
>
> We see that the basic TokenLearner + ViViT L/16 performs better than ViViT, while spending ~1/2 times FLOPS. Our TokenLearner + L/10 accuracy, 85.4, is to our knowledge, the best known accuracy on Kinetics 400.
>
> ### What is the accuracy of a network that exclusively uses this blocks ones?
>
> We experimented with inserting TokenLearner at different locations within ViViT: We found that it performed best when inserting it around the middle of the network. We believe this is because TokenLearner requires the inputs to be sufficiently "high-level" in order to summarize it into a small number of tokens. Intuitively, we need some initial layers of unmodified encoder blocks, so the tokens can summarize the relevant information first. We save more compute the earlier we insert TokenLearner into the network.
>
> ### Computational costs of the methods?
>
> We will revise the tables in the paper to include FLOPS. For instance, Table 2 with AViD results will be updated as follows:
>
> \\begin{array} {l|c|c}
> \\text{Method} & \\text{Accuracy} & \\text{total GFLOPS} \\\\
> \\hline
> \\text{I3D} & 46.5 & 108 \times N/A\\\\
> \\text{(2+1)D ResNet-50} & 46.7 & 152 \times 115\\\\
> \\text{SlowFast-50 8x8}  & 50.2 & 65.7 \times 30 \\\\
> \\text{SlowFast-101 16x4}  & 50.8 & 213 \times 30\\\\
> \\hline
> \\text{Backbone (X(2+1)D-M)} & 48.6 & 532 \times 1\\\\
> \\text{X(2+1)D-M w/ Space+Time transformer} & 50.6& 493 \times 1 \\\\
> \\text{Ours} & 53.6 & 487 \times 1\\\\
> \\end{array}
>
> Observe how our models outperform prior work in terms of both accuracy and FLOPs simultaneously.
>
> Note that our models follow a slightly different inference method compared to SlowFast. We apply our model in a fully-convolutional fashion over all the frames and average the result. SlowFast, on the other hand, averages results over 30 separate spatio-temporal views of the input video clip.
>
> ### Data augmentation used in the experiments
>
> For AVID and Charades, the augmentations used are random temporal and spatial crops during training. The augmentation used for Kinetics are: random temporal cropping, random spatial cropping, random rescaling image, and random colour jitter as was done in ViViT.
>
> We will include the full training details (e.g., learning rate, # of iterations, …) for each experiment in the final version of the paper. We will also release the code together with the final version of the paper.

---

### Official Review · Reviewer_TXug · 2021-07-16

**Rating:** 6
**Confidence:** 4

**Summary:**

The paper proposes a transformer-based neural network architecture for extracting video representations. Past approaches discretize entire videos into 3D (RGB-T) chunks as input 'tokens' to transformers. Instead in this work, an additional operation is proposed to extract a smaller number of tokens per frame. This leads to a more computationally efficient architecture that outperforms existing methods on the task of video classification.

**Limitations And Societal Impact:**

Yes, the authors have addressed limitations and potential impact thoroughly.


**Main Review:**

# Strengths

- The proposed idea of extracting limited number of tokens per frame is intuitive and easy to follow.

- The attention mechanism proposed to extract fixed number of tokens is novel. Due to its simplicity it could be easily adopted in different computer vision applications. Considering the fact that transformers bring heavy computational cost in numerous domains, this idea could potentially be impactful.

- Experimental results demonstrate that the model outperforms existing architectures for video classification on a wide range of multiple datasets. The ablative studies also show that the individual components used, VecT, TokenLearner and the Fusion operator, all contribute towards improve performance of the model.

# Concerns

## Writing
- The abstract and introduction are not easy to follow. For example, the abstract begins by talking about space-time "tokens". To the best of my knowledge, this is not a term used commonly in computer vision.
"The proposed approach is designed with the intention to allow the tokenizer to adaptively react to input video frames containing diverse visual content, and then to have the vector transformer and subsequent modules learn the underlying spatio-temporal interactions and long-range dependencies in video inputs."
This sentence is extremely difficult to follow without any context provided.
- Line 37: "Further, with these adaptively learned tokens, we apply a vector attention between key-query pairs." So far keys and queries have not been introduced and it seems like the reader is expected to know what "vector attention" is.
- Line 38: "This can be thought as a version of Transformer in which the number of heads is the same as channels, allowing us to learn a different attention matrix for each channel." This also doesn't make sense without any additional information.

## Approach Details
- Line 91, why is $|Z| = S$ and not $ST$?
- Line 99, "the attention tensor is multiplied with the value matrix". Here the attention tensor is $ST \times ST \times C$ and the value matrix is $ST \times C$. So what kind of multiplication are you performing here?
- Line 122-130, since your model is built on the X3D architecture, it would be useful to explain it in more detail here. For readers who are not well-versed with this literature, the rest of the description doesn't make sense.

## Efficiency and Performance
- Comparing Row 4 and Row 7 of Table 3, it looks like your model is significantly more inefficient while only providing a ~1% gain in performance. In this light, what is the significance of the proposed architecture design choices?
- Line 202-Line 206, it seems like removing the learned tokenizer and replacing it with a predefined discrete grid only causes a drop in performance of ~0.8%. Again, here it doesn't seem like the TokenLearner is contributing much compared to this naive method.

**Time Spent Reviewing:**

3 hours

---

> ### Author Response · Authors · 2021-08-10
> **Thank you for the comments. Please find our answers below.**
>
> ### Writing
>
> Thanks for pointing this out. In the introduction and abstract, we will define “tokens” from the beginning and more clearly define “vector attention” up front. We will also cover necessary background and terminology before using them.
>
> ### Approach details
>
> - *Line 91, why is |Z|=S and not ST?* This is a typo and we will fix it. Thanks.
>
> - *What kind of multiplication are we performing in Line 99?* It is a matrix multiplication per channel, implemented by treating the channel axis as a batch axis. For each channel, it multiplies an attention matrix of size STxST and a value matrix of STx1. Concatenating the results generates an output matrix of size ST×C. This channel-wise matrix multiplication results in the same set of operations as in Eq. 2.
>
> - *X3D details?* Thank you for the suggestion. We will include the details in the paper.
>
> ### Efficiency and Performance
>
> **Table 3 comparison**
> We apologize. There was an error in Table 3 FLOPS measure that caused extra computations for transformers, which we correct as below. In addition, we added two entries, (1) Conv2D + Spatial + Temporal transformer and (2) TokenLearner-only, which would allow more direct comparison between FLOPS of Spatial + Temporal transformer vs. TokenLearner.
>
> \\begin{array} {l|c|c|c}
> \\text{Method} & \\text{Accuracy} & \\text{GFLOPS} & \\text{params} \\\\
> \\hline
> \\text{Conv2D + Conv1D} & 56.6 & 18.3 & 2.24M\\\\
> \\text{Conv2D + MLPMixer} & 57.0 & 13.8 & 2.06M\\\\
> \\text{Conv2D + Temporal transformer} & 58.4 & 16.5 & 1.98M\\\\
> \\text{Spatial + Temporal transformer} & 58.5 & 5.5 & 0.59M\\\\
> \\text{Conv2D + Spatial + Temporal transformer} & 58.0 & 19.2 & 2.27M\\\\
> \\text{Ours (TokenLearner)} & 58.5 & 3.4 & 0.81M\\\\
> \\text{Ours (SpatialT + TokenLearner)} & 58.7 & 6.2 & 1.11M\\\\
> \\text{Ours (Conv2D + TokenLearner)} & 59.6 & 17.2 & 2.49M\\\\
> \\end{array}
>
> If we compare Spatial + Temporal transformer vs. TokenLearner-only, they get similar accuracies but TokenLearner has fewer FLOPS. If we compare Conv2D + Spatial + Temporal transformer vs. Conv2D + TokenLearner, TokenLearner was able to better benefit from Conv2D while Spatial+Temporal transformer failed to take advantage of it, which probably is due to the spatial transformer mixing token locations.
>
> **The difference between TokenLearner vs. Pooling with a discrete grid is only 0.8%**
>
> This is a good observation. In order to further investigate, we tested TokenLearner and the grid-based pooling-baseline on larger datasets, AViD and Kinetics400. The limitation of Charades is that it has less training data compared to AViD and Kinetics, making the spatial token learning more challenging when trained from scratch.
>
> On AViD (which is the largest dataset we used), while spending similar FLOPS, the accuracies of TokenLearner vs. pooling-baseline are 53.6 vs 51.3. On Kinetics400, TokenLearner vs. pooling were 83.5 vs. 82.3. Similar to the Charades experiment, we made the pooling-baseline to pool a smaller number of tokens (8 for AViD and 16 for Kinetics) at the same location where TokenLearner is introduced.
>
> TokenLearner is therefore effective on all three datasets, particularly on the larger AViD and Kinetics datasets

---

### Official Review · Reviewer_EDW8 · 2021-07-20

**Rating:** 7
**Confidence:** 3

**Summary:**

This paper introduces a method of adaptive space-time tokenization (TokenLearner) for video representation learning. Instead of using fixed/pre-trained tokens for videos, TokenLearner aims to generate multiple spatial attention maps for the architecture to better focus on regions with rich information, thus obtaining better tokens. The overall architecture mimics a combination of X(2+1)D and Bottleneck Transformer while inserting the TokenLearner module. The authors conduct experiments on Charades and AViD datasets, and the proposed method outperforms previous state-of-the-art models. The authors have also carried out ablation study on their design choices.

**Ethical Concerns:**

I don't find significant ethical concerns.

**Limitations And Societal Impact:**

The authors have addressed potential societal impact in the section "Broader Impact".

**Main Review:**

Overall, the work is interesting, inspiring and promising. The proposed method is well motivated and investigated. The experimental results are impressive.

**Originality**: To the best of my knowledge, the method is novel.

**Clarity**: The paper is mostly well-written and easy to follow. Clarification on the method details is welcomed.

**Significance**: The experimental results are promising, as TokenLearner achieves state-of-the-art performance on both datasets of Charades and AViD. It will be more beneficial to the community if the codebase and models can be released.

**Strengths**:

1. The method is well motivated. Video data are of large size and an efficient and effective model should adaptively focus on regions with rich information. The authors propose to enable such adaptive attention in the tokenization of videos.
2. The experimental results are impressive. On Charades, without pre-training, the proposed method outperforms previous state-of-the-art methods. On AViD, the proposed method outperforms prior work by a significant margin.
3. Exhaustive ablation studies are conducted, including different space-time transformers, different model components, and alternative designs to TokenLearner.
4. Visualization of the spatial attention maps looks convincing.

**Weaknesses and Questions**:

1. The function $\alpha_i(\cdot)$ is very light-weighted as it is only a single convolutional layer (or a MLP layer) followed by a sigmoid activation (L79-80). However, I wonder if a single learnable layer is sufficient for the job of generating various and diversed attention maps. Have you tried larger models for $\alpha_i(\cdot)$?

2. My understanding is that the adaptive tokenization only happens on the spatial dimension, as there are only spatial attention maps learned in the tokenization. Have you tried similar approach on the time dimension?

3. Computation in the section 2.2 is not very clear to me. What is the shape of $f_q(z_i)$, $f_k(z_j)$ and $\gamma(f_q(z_i) \odot f_k(z_j))$? More mathematical details are needed in L91-100.

4. How many crops are used in Charades evaluation? Prior work shows that the number of crops affects the final performance greatly, and I wonder if consistent hyperparameters are used in prior work and this paper.

5. On Charades, the performance gain due to TokenLearner seems marginal (shown by comparing the last two rows in Table 1, mAP 58.5 vs 59.6). However, TokenLearner seems more helpful on AViD (shown by comparing the last two rows in Table 2, accuracy 50.1 vs 53.6). Any thoughts on why and what TokenLearner is actually helping with?

6. Typos.
- L63: "TokenLearner"
- L91: shouldn't it be $|Z| = ST$?

**Time Spent Reviewing:**

6

---

> ### Author Response · Authors · 2021-08-10
> **Thank you for the comments. Please find our answers below.**
>
> ### Larger models for the function $\alpha()$?
>
> Yes, great point, more advanced versions can be explored. We tried using 4 conv layers (with gelu in between), which we found to work better particularly when TokenLearner is added earlier in the model, but opted for a faster version in the final version. We also tried having a small resnet-18 as our $\alpha()$, but it did not make much difference while spending many more FLOPS.
>
> ### What is the shape of fq(zi), fk(zj) and γ(fq(zi)⊙fk(zj))?
>
> $f_q(z_i)$ and $f_k(z_j)$ are the embedding functions taking inputs with the shape of 1xC and generating the outputs with the shape 1xC'. C' is an intermediate channel size as in other transformers, which is 64 in our case. $f_q(z_i) \odot f_k(z_j)$ is also of shape 1xC' as it is the element-wise multiplication between the two vectors. γ maps the channel dim back to 1xC.
> We will clarify this further in the paper.
>
> ### How many crops are used in Charades evaluation?
> We used the same setting used in [27] for Charades. Specifically, we used two temporal views (i.e., crops) per video. Charades videos on average are 30 seconds long. Each input to our model is 64 frames with 6fps (roughly 11 seconds), making two views sufficient to roughly cover the entire video. No spatial cropping was used. We will clarify this further in the paper.
>
> ### Why is TokenLearner more beneficially on AViD compared to Charades?
> Yes, this is a good discussion point. We believe it is due to the fact that AViD is a much larger dataset with many more training annotations. TokenLearner requires training of the attention function $\alpha()$, and more data in AViD facilitates the optimization of TokenLearner. We also believe that AViD is a more challenging and less structured dataset with more diverse content, making the spatial attention ability of TokenLearner more necessary. This explanation is also consistent with our results on the Kinetics dataset, which is also larger and more diverse than Charades.
>
> ### Typos
>
> We will revise the paper and fix typos. Thank you.

---

### Decision · Program_Chairs · 2021-09-27

**Decision:**

Accept (Poster)

**Comment:**

After the rebuttal period all reviewers rate this paper as being past the threshold for acceptance.
The authors did a good job of addressing questions of novelty during the rebuttal and additional experiments were well received by reviewers. The authors have promised to clean and polish some parts of the manuscript flagged by reviewers - please do this.

The AC recommends acceptance